# A Systematic Review and Meta-Analysis of Serologic Response following Coronavirus Disease 2019 (COVID-19) Vaccination in Solid Organ Transplant Recipients

**DOI:** 10.3390/v14081822

**Published:** 2022-08-19

**Authors:** Atsushi Sakuraba, Alexander Luna, Dejan Micic

**Affiliations:** Division of Gastroenterology, Hepatology and Nutrition, Department of Internal Medicine, The University of Chicago Medicine, 5841 S. Maryland Ave. MC 4076, Chicago, IL 60637, USA

**Keywords:** COVID-19, vaccine, outcomes, meta-analysis, solid organ transplant, immunocompromised, kidney transplant, heart transplant, lung transplant

## Abstract

Solid organ transplant (SOT) recipients are at greater risk of coronavirus disease 2019 (COVID-19) and have attenuated response to vaccinations. In the present meta-analysis, we aimed to evaluate the serologic response to the COVID-19 vaccine in SOT recipients. A search of electronic databases was conducted to identify SOT studies that reported the serologic response to COVID-19 vaccination. We analyzed 44 observational studies including 6158 SOT recipients. Most studies were on mRNA vaccination (mRNA-1273 or BNT162b2). After a single and two doses of vaccine, serologic response rates were 8.6% (95% CI 6.8–11.0) and 34.2% (95% CI 30.1–38.7), respectively. Compared to controls, response rates were lower after a single and two doses of vaccine (OR 0.0049 [95% CI 0.0021–0.012] and 0.0057 [95% CI 0.0030–0.011], respectively). A third dose improved the rate to 65.6% (95% CI 60.4–70.2), but in a subset of patients who had not achieved a response after two doses, it remained low at 35.7% (95% CI 21.2–53.3). In summary, only a small proportion of SOT recipients achieved serologic response to the COVID-19 mRNA vaccine, and that even the third dose had an insufficient response. Alternative strategies for prophylaxis in SOT patients need to be developed. **Key Contribution:** In this meta-analysis that included 6158 solid organ transplant recipients, the serologic response to the COVID-19 vaccine was extremely low after one (8.6%) and two doses (34.2%). The third dose of the vaccine improved the rate only to 66%, and in the subset of patients who had not achieved a response after two doses, it remained low at 36%. The results of our study suggest that a significant proportion of solid organ transplant recipients are unable to achieve a sufficient serologic response after completing not only the two series of vaccination but also the third booster dose. There is an urgent need to develop strategies for prophylaxis including modified vaccine schedules or the use of monoclonal antibodies in this vulnerable patient population.

## 1. Introduction

Coronavirus disease 2019 (COVID-19) caused by severe acute respiratory syndrome coronavirus 2 (SARS-CoV-2) has led to a global health emergency [1]. It has been reported that older individuals, patients with pre-existing comorbidities, and those that are immunosuppressed are at greater risk of COVID-19 and experiencing severe pneumonia [2,3,4]. Patients who have undergone organ transplants require lifelong immunosuppressive therapy. Long-term maintenance immunosuppression including the combinatory use of calcineurin inhibitors, antimetabolites, glucocorticoids (GCs), monoclonal antibodies, and inhibitors of T cell activation remains the mainstay in improving graft survival in transplant recipients [5,6]. In the United States, 39,000 organ transplants were performed in 2020 [7]. Renal transplants were the most commonly performed, followed by liver, heart, and lung transplantation [7]. Furthermore, more than 100,000 patients are on the national transplant waiting lists. Patients with solid organ transplants (SOT) are at risk for various infections because the immunosuppressive agents dramatically reduce immunity against viruses, bacteria, and other pathogens [8,9]. During periods of a pandemic, the protection of the safety or health of SOT patients is an important priority [10].

Salto-Alejandre et al., demonstrated that old age and a shorter interval between transplant and COVID-19 diagnosis showed an association with poor COVID-19 outcomes in SOT patients with a case fatality rate greater than 20% [4]. These reports suggest that there is a need for effective prophylaxis in SOT recipients. Due to the lack of efficacious therapies for COVID-19, vaccinations that could prevent infections are of importance in lowering the mortality risk [11]. COVID-19 vaccines were developed and a large number of studies demonstrated that mRNA COVID-19 vaccinations are efficacious and safe [12]. Patients who were on immunosuppressive drugs were excluded from initial vaccination trials, so the effectiveness of COVID-19 vaccinations in transplant recipients remains unclear [13]. Guideline recommendations are that transplant recipients should receive COVID-19 vaccination unless there is any contraindication. Studies of other infectious conditions have shown that SOT recipients have diminished humoral response to vaccinations [14]. Boyarsky et al., showed that kidney or liver transplant patients had decreased immunogenicity to mRNA-1273 and BNT162b2 vaccinations [15]. Most studies assessing the efficacy of COVID-19 vaccinations in transplant recipients are of limited sample size [16,17].

In order to improve clinical care and protect this vulnerable patient population, it is important to understand the efficacy of COVID-19 vaccinations in transplant recipients. We aimed to integrate data from various studies to evaluate the serologic response to COVID-19 vaccines in SOT patients.

## 2. Methodology

### 2.1. Search Strategy of Selecting Studies

We conducted the present meta-analysis according to a protocol that is in accordance with the PRISMA guideline [18]. We have submitted the protocol to the International Prospective Register of Systematic Reviews (PROSPERO) (pending approval) [19]. PubMed/MEDLINE, EMBASE, and medRxiv were electronically searched on 1 August 2021. 

Observational studies that reported the serologic response to COVID-19 vaccines in SOT recipients were included. No restrictions for country, language, age, or type of vaccine of the study were imposed. Two members (AS and AL) screened each potential study in an independent manner to assess the eligibility. When any issues or disagreements were found, they were resolved after discussion. We used the following terms to search for eligible studies: “SARS-CoV-2”, “COVID-19”, “vaccine”, “transplant”, “transplantation”, “liver”, “kidney”, “heart”, “lung”, “pancreas”, “multi-organ”, and “immunosuppressed”. We also searched bibliographies of eligible articles for any potential references. Manuscripts that were published in non-English language were translated if necessary. We excluded single case reports and studies that only reported adverse effects of COVID-19 vaccines. We contacted the corresponding author of studies to obtain missing data when necessary. Figure 1 shows the search strategy.

### 2.2. Extraction of Data and Assessment of Quality

The data of each eligible study were independently extracted in duplicate by the authors (AS and AL). Extracted data included study characteristics such as name of authors, publication year, country, administered vaccine, duration, sample size, transplanted organ, treatment, characteristics of patients, and results of serologic tests. The diseases were categorized into the following groups according to the transplanted organ: (1) kidney, (2) heart/lung, and (3) liver. Studies that reported outcomes in various organ transplant recipients without distinction were classified as (4) mixed. 

The risk of bias of the studies was assessed by the Joanna Briggs Institute Critical Appraisal Checklist [20,21]. The quality of evidence obtained from the present meta-analysis was rated by the Grades of Recommendation, Assessment, Development and Evaluation (GRADE) [22].

### 2.3. Assessment of Outcomes

Serologic response rate to COVID-19 vaccine was the primary outcome of the present meta-analysis. We separately assessed the rate of response following the first, second, and third vaccination. The rate of serologic response in transplant recipients compared to control population was the secondary outcome. The number of patients achieving antibody levels above the cut-off of the test method used in each study divided by the total number of patients was defined as the rate of serologic response [23]. A majority of studies used one of the three commercially available antibody tests (Roche, DiaSorin, and Abbott), so applying a common cut-off value between studies was not possible; however, they all have excellent sensitivity (98–100%) [21,24]. When response was tested at different timing, the time point closest to 4 weeks following the vaccine was chosen [21].

Meta-regression and subgroup analyses based on transplanted organs (kidney, heart/lung, liver, mixed group), patient’s age, and proportion of patients on immunosuppressive medications were performed. Immunosuppressive medications such as calcineurin inhibitors, mycophenolate, antimetabolites, monoclonal antibodies, and inhibitors of T cell activation were included, but were not assessed separately because a majority of studies did not report outcomes separately according to treatment.

### 2.4. Statistical Analysis

A meta-analysis of the serologic response rate to COVID-19 vaccine was undertaken by random effects model [21]. Heterogeneity across included studies was analyzed by *I*^2^ statistic. Heterogeneity was defined as low, moderate, and high when *I*^2^ value was <25%, 25–75%, and >75%, respectively [25]. Heterogeneity was also evaluated by Cochran’s Q-statistics (*p* < 0.10) [26]. Publication bias was analyzed with Begg’s and Egger’s tests. When more than 3 studies were included in the meta-analysis, funnel plots were created to assess asymmetry [27,28]. Subgroup analyses and random effects meta-regression models were used to evaluate the contribution of each risk factor to the response to vaccination [21]. When performing multivariate meta-regression, all variables included in the univariate meta-regression were included as they were all considered clinically important.

Preprint studies were included because they are a substantial part of the available COVID-19 literature [23]. However, we conducted sensitivity analyses by excluding preprint studies because they still lack peer-review [29]. One study included exclusively a non-mRNA vaccine (Ad26.COV2.S), so it was analyzed separately. One study removed analyses that were undertaken to assess whether the results of the meta-analyses were biased by a single study [21].

We used Comprehensive Meta-Analysis Software (version 3; Biostat, Englewood, NJ, USA) for all statistical analyses. We used a two-sided *p*-value of 0.05 for significance except for Q-statistics.

### 2.5. Data Sharing and Access

Data of the present study will be made available when requested from the corresponding author. All coauthors of the present study had access to the data of the study and approved final version of the manuscript.

## 3. Results

### 3.1. Study Characteristics

As shown in Figure 1, 1941 studies were identified through the literature search. We excluded 1833 studies after screening of title and abstract. We assessed 108 studies for eligibility and found 44 studies (6158 patients) that met the eligibility criteria. Twenty-two studies included only patients with kidney transplant, seven included patients with heart and/or lung transplant, and one included patients with liver transplant (Table 1). Fourteen studies included patients with various SOTs (mixed group), which included kidney, heart/lung, liver, and pancreas, but did not report outcomes separately. Twenty-six and five studies included patients vaccinated with only BNT162b2 and mRNA-1273, respectively. Eight studies included patients vaccinated with either BNT162b2 or mRNA-1273. One study only used AD26.COV2.S. One study used ChAdOx1 nCoV-19 as a third dose in patients who previously received two series of BNT162b2. Another study used AD26.COV2.S in patients who previously received two series of BNT162b2 or mRNA-1273.

Twenty-one and thirty-five studies were eligible for evaluation of serologic response after one and two doses, respectively. Four and fifteen studies compared serologic response after one or two doses of COVID-19 vaccination to controls with no history of transplant. Twelve studies reported outcomes in patients who received a third vaccine dose [16,17,30,31,32,33,34,35,36,37]. Most studies measured outcomes 3–5 weeks following the vaccine (Table 1). The summary of characteristics of the studies that were included in the meta-analysis is shown in Table 1. We evaluated the risk of bias in the studies with the Joanna Briggs Institute Critical Appraisal Checklist (Appendix A). Most of the studies were of medium–high quality.

### 3.2. Serologic Response following a Single Dose of COVID-19 Vaccination

There were 21 studies that evaluated the serologic response following the first vaccination in transplant recipients (20 mRNA vaccine and one AD26.COV2.S vaccine). The serologic response following a single dose of mRNA vaccine was extremely low at 8.6% (95% confidence interval [CI] 6.8–11.0) (Figure 2A). Subgroup analysis demonstrated that the rates were 7.0% (95% CI 3.3–14.0), 6.6% (95% CI 4.3–9.9), and 10.7% (95% CI 7.8–14.7) in heart or lung transplant recipients, kidney transplant recipients, and mixed transplant recipients, respectively. The one study that reported a serologic response following a single dose of AD26.COV2.S vaccination showed a rate of 16.7% (Appendix A).

Heterogeneity was present (*I*^2^ = 74.2%) in the meta-analysis of mRNA vaccines possibly due to the differences in sample size and reported rates of the studies. Multivariate meta-regression showed that the proportion of patients on GCs (regression coefficient −0.029, 95% CI −0.052–0.0061, *p* = 0.013) was a source of heterogeneity. Studies that used both types of mRNA vaccines showed a greater response rate compared to studies that only included BNT162b2 vaccine (coefficient 1.08, 95% CI 0.20–1.96), *p* = 0.016) (Appendix A).

Visual evaluation of the funnel plot of the mRNA vaccine studies showed no asymmetry; however, we found publication bias with Egger’s test (*p* < 0.001), but not with Begg’s test (*p* = 0.36) (Appendix A).

We performed sensitivity analyses to evaluate whether any single study influenced the outcomes (Appendix A). The results were unchanged when each study was removed one at a time from the meta-analyses. Specifically, the removal of the one preprint study showed similar results [37]. Subgroup analysis stratifying based on the type of the vaccination demonstrated that the serologic response rates were 5.4%, 11.8%, and 10.7% in studies that used BNT162b2, mRNA-1273, or both types of vaccines, respectively (Appendix A).

### 3.3. Serologic Response following Two Doses of COVID-19 Vaccination

Thirty-five studies were available for the assessment of the serologic response following two doses of mRNA vaccination. There were no studies that used non-mRNA vaccines. The pooled serologic response rate was 34.2% (95% CI 30.1–38.7) (Figure 2B). Subgroup analysis demonstrated that the rates were 22.9% (95% CI 11.1–45.9), 28.3% (95% CI 22.6–34.8), and 35.5% (95% CI 29.4–44.2) in heart/lung transplant recipients, kidney transplant recipients, and mixed transplant group, respectively. The one liver transplant study reported a rate of 47.5%.

Heterogeneity was present (*I*^2^ = 88.9%) which was possibly due to the differences in the sample size and the response rates of the included studies. Multivariate meta-regression demonstrated that older age (regression coefficient −0.10, 95% CI −0.19–0.020), *p* = 0.016) and GC use (regression coefficient −0.014, 95% CI −0.025–0.0024), *p* = 0.017) were significant sources of heterogeneity. (Appendix A).

Visual evaluation of the funnel plot showed no asymmetry; however, we found publication bias with Egger’s and Begg’s tests (*p* < 0.001 and *p* = 0.0038, respectively) (Appendix A).

We performed sensitivity analyses to evaluate whether any single study influenced the outcomes (Appendix A). The results were unchanged when each study was removed one at a time from the meta-analyses. Removal of the three preprint studies showed similar results (Appendix A). Subgroup analysis stratifying based on the type of vaccine showed that the rates were 28.9%, 26.6%, and 40.6% in studies that used BNT162b2, mRNA-1273, or both types of vaccines, respectively (Appendix A).

A few studies that reported antibody titers or concentrations showed greater than ten-fold lower values in transplant recipients (Table 1).

### 3.4. Serologic Response following a Single Dose of COVID-19 Vaccination Compared to Controls

There were four studies that included control patients after one dose of the vaccine. Meta-analysis demonstrated that transplant recipients were less likely to achieve a serologic response compared to controls (6.3% 94.5%; odds ratio [OR] 0.0049, 95% CI 0.0021–0.012, *p* < 0.001) (Figure 3A). Subgroup analysis demonstrated that both heart and/or lung and kidney transplant recipients achieved statistically significantly lower rates of response compared to control patients (4.0% vs. 98.0%; OR 0.0009, 95% CI 0.0001–0.0097, *p* < 0.001 and 6.7% vs. 92.8%; OR 0.0049, 95% CI 0.0021–0.012, *p* < 0.001, respectively).

Heterogeneity was absent (*I*^2^ = 16.0%). Visual evaluation of the funnel plot showed no asymmetry, and no publication bias was found (Begg’s *p* = 0.73, Egger’s *p* = 0.51) (Appendix A).

### 3.5. Serologic Response following Two Doses of COVID-19 Vaccination Compared to Controls

There were 15 studies that included control patients after two doses of vaccine. Meta-analysis demonstrated that transplant recipients were less likely to achieve a serologic response compared to controls (30.8% vs. 99.4%; OR 0.0057 (95% CI 0.0030–0.011), *p* < 0.001) (Figure 3B). Subgroup analysis demonstrated that heart and/or lung and kidney transplant recipients as well as the mixed transplant group achieved statistically significantly lower response rates compared to controls (12.1% vs. 99.0%; OR 0.0024 [95% CI 0.0007–0.0084], 31.8% vs. 99.3%; OR 0.0063 [95% CI 0.0025–0.016], 35.5% vs. 100%; OR 0.011 [95% CI 0.0025–0.045], respectively, and all *p* < 0.001).

There was no heterogeneity (*I*^2^ = 0%). Visual evaluation of the funnel plot showed no asymmetry, and no publication bias was found with Begg’s and Egger’s tests (*p* = 0.20, *p* = 0.47) (Appendix A).

### 3.6. Serologic Response following Three Doses of COVID-19 Vaccination

Ten studies reported the serologic response following a third dose of COVID-19 vaccination. Five studies used the same mRNA vaccine for all three doses (BNT162b2 or mRNA-1273) whereas three studies used a different mRNA vaccine for the third dose. Two studies used mRNA vaccines as the first two series followed by Ad26.COV2.S as the third dose. As can be seen in Figure 4A, the rate of serologic response was 65.5% (95% CI 60.4–70.2), which was nearly two-fold greater than the rate after two doses. Subgroup analysis demonstrated that the rates were 66.7% (95% CI 56.7–75.4), 66.9% (95% CI 60.1–73.2), and 59.9% (95% CI 47.9–70.8) in heart and/or lung transplant recipients, kidney transplant recipients, and mixed group, respectively. The rate was lower with Ad26.COV2.S as the third dose compared to using the same or different combination of mRNA vaccines (41.0% (95% CI 24.2–60.2%) vs. 66.1% (95% CI 62–69.9%)–70.1% (95% CI 43.7–87.6%), respectively) (Appendix A). The results were unchanged when each study was removed one at a time from the meta-analyses (Appendix A).

Heterogeneity was moderate (*I*^2^ = 44.7%) possibly due to the differences in the sample size and reported rates of the studies. Visual evaluation of the funnel plot showed no asymmetry, and no publication bias was found with Begg’s and Egger’s tests (*p* = 0.37, *p* = 0.19) (Appendix A).

### 3.7. Serologic Response following Three Doses of COVID-19 Vaccination in Non-Responders after Two Doses

There were six studies that reported the serologic response after a third dose of the COVID-19 vaccine in patients who did not show a response following two vaccination. All studies only included kidney transplant recipients. Four studies used the same mRNA vaccine for all three doses (BNT162b2 or mRNA-1273) whereas one study each used BNT162b2 as the first two series followed by AZD1222 or mRNA-1273 as the third dose. As can be seen in Figure 4B, the serologic response rate was 35.7% (95% CI 21.2–53.3), which was half of that seen among studies not restricted to two-dose non-responders (Figure 4A). Subgroup analysis demonstrated that the serologic response rates were 54.5% (95% CI 26.8–79.7) and 60.0% (95% CI 29.7–84.2) when AZD122 or mRNA-1273 were administered as the third dose following two doses of BNT162b2, but 27.9% (95% CI 13.3–49.4) when the same mRNA vaccines were used for all three doses (Appendix A). It should be noted that the number of studies that used a different type of the third vaccine was one each. The results were unchanged when each study was removed one at a time from the meta-analyses (Appendix A).

Heterogeneity was present (*I*^2^ = 82.1%) possibly due to the differences in the sample size and reported rates of the studies. Visual evaluation of the funnel plot showed no asymmetry, and no publication bias was found with Begg’s and Egger’s tests (*p* = 0.71, *p* = 0.36) (Appendix A).

### 3.8. Quality of Evidence Assessed by GRADE

The quality of evidence of this analysis was considered low because the data were derived mostly from observational or cohort studies (Appendix A).

## 4. Discussion

Our meta-analysis of the serologic response to the COVID-19 vaccine in SOT recipients showed that less than 10% of transplant recipients seroconverted following one dose of COVID-19 vaccination. The rate improved to 34% following the second dose. In comparison to controls, the seroconversion rates were lower following both doses. A third booster shot, which has recently been approved, improved the rate to 66%, but in a subset of patients who had not achieved a response after two doses, it remained low at 36%. The results of our study suggest that there is a need for a more effective prophylaxis strategy in SOT patients.

Patients with a history of SOT are at higher risk for COVID-19 and its mortality [38]. Transplant recipients are immunocompromised due to the long-term use of immunosuppressive medications. Immunosuppressive drugs may influence the functions of B and T cells that lower the response to vaccinations [21,23,39]. Furthermore, transplant recipients are generally older and may carry comorbidities that increase the risk of COVID-19.

The study by Shroti et al., reported that 96–99% of patients in the general community seroconverted following a single or two series of ChAdOx1 nCoV-19 or BNT162b2 vaccines. In their study, elderly individuals and those with cardiovascular disease, diabetes, or cancer were less likely to achieve antibody response [40]. In our study, the serologic response rates of SOT patients following one or two doses of COVID-19 vaccination was 9% and 34%, respectively, which were lower than those reported in a meta-analysis of immune mediated inflammatory disease patients receiving biologics [23]. When compared to controls, the odds of seroconversion among transplant recipients were lower after both doses. The proportion of patients on GC therapy was associated with lower response following one dose whereas advanced age was associated with lower response in addition to GC use following two doses. This suggests that additional patient and treatment factors may be associated with a weakened vaccine response in transplant recipients, which warrants further investigation.

The low serologic response rates to the two-dose mRNA vaccination strategy in transplant recipients led us to investigate the response to the booster (third) vaccination. The response rate after the third dose rose to 65%, but was still suboptimal. Furthermore, among a subset of patients who failed to seroconvert after two doses, merely a third seroconverted after the third dose. AD26.COV2.S mounted a weaker serologic response, but AZD1222 or mRNA-1273 induced a stronger response in subgroup analyses, so studies of different hybrid regimens or doses are warranted [41,42]. Future studies will need to assess strategies for prophylaxis including the use of monoclonal Antibody (mAb)-based immune prophylaxis as a substitute or as a complement to vaccines in SOT recipients.

## 5. Limitations

Although it has been more than a year since the first vaccine was approved against COVID-19, available reports on transplant recipients were rather limited and mostly consisted of studies of a small sample size. There are currently nine COVID-19 vaccinations available for use globally, but a majority of the studies included in our meta-analysis were of either mRNA-1273 or BNT162b2, and a small number of studies were of AD26.COV2.S or AZD-1222/ChAdOx1 nCoV-19. Our primary outcome was a humoral response to vaccines; however, we could not evaluate the difference in T cell-mediated immune response due to the of data. Recent studies have reported that levels of antibodies can be predictive of the risk of infections in healthy individuals and that those who are immunocompromised were at greater risk of post-vaccination infections, so we consider that our meta-analysis focusing on serologic response provides meaningful information [43,44]. Further studies assessing whether the incomplete serologic response to vaccination would prevent symptomatic or severe COVID-19 in SOT patients are also warranted [45,46]. Due to the limited data, we were unable to undertake subgroup analyses according to different immunosuppressive therapies. Furthermore, the studies included in our meta-analysis were somewhat heterogeneous regarding transplanted organ(s), size of the sample, treatment, timing, and type of test used to measure antibody levels. A majority of studies included in our analysis used either the antibody test marketed by Abbott, DiaSorin, or Roche. They are all known to have a sensitivity of 98 to 100% [24]. Consequently, the results of antibody testing have an excellent correlation in regard to seroprevalence. Recent reports have shown declining antibody levels after 3–8 weeks of second vaccination in patients receiving ChAdOx1 nCoV-19 or BNT162b2 [47], so further research assessing the waning of antibodies in immunocompromised and transplant patients is warranted.

## 6. Conclusions

We have demonstrated that the rate of seroconversion to vaccinations against COVID-19 in SOT recipients was only 9% and 34% following the first and second dose. The rate improved to 66% following the third booster injection. However, in a subset of patients who had not achieved a response after two doses only a third achieved a response with the third dose. The results of our study suggest the urgent need for an improved prophylaxis strategy including the use of mAb-based immune prophylaxis as a substitute or as a complement to vaccines in this vulnerable patient population and that they need to continue following safety measures.

## Figures and Tables

**Figure 1 viruses-14-01822-f001:**
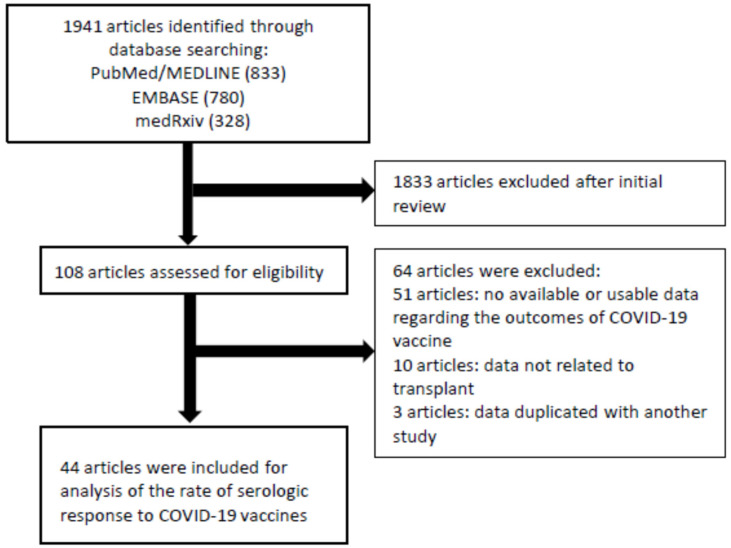
Flowchart of the assessment of the studies identified in the meta-analysis.

**Figure 2 viruses-14-01822-f002:**
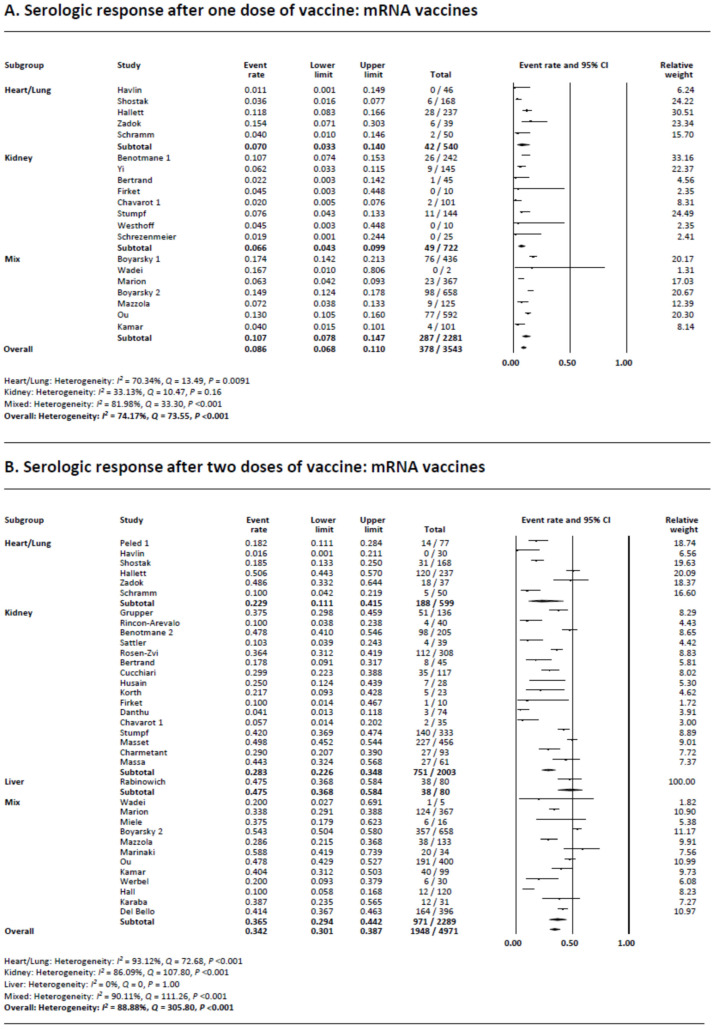
(**A**) Meta-analysis of serological response after one dose of vaccine: mRNA vaccines. (**B**) Meta-analysis of serological response after two doses of vaccine: mRNA vaccines.

**Figure 3 viruses-14-01822-f003:**
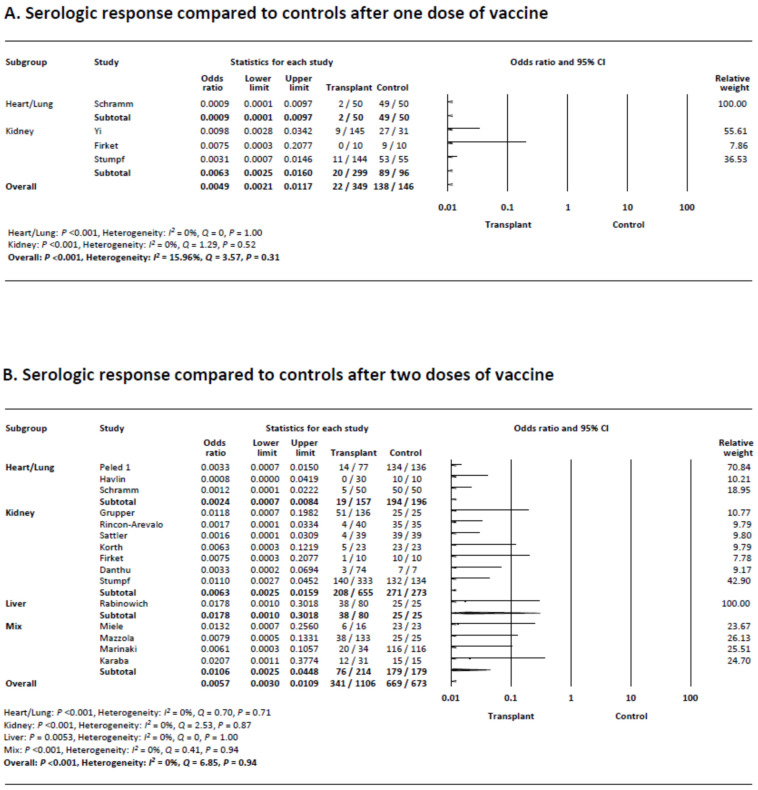
(**A**) Meta-analysis of serological response compared to controls after one dose of vaccine. (**B**) Meta-analysis of serological response compared to controls after two doses of vaccine.

**Figure 4 viruses-14-01822-f004:**
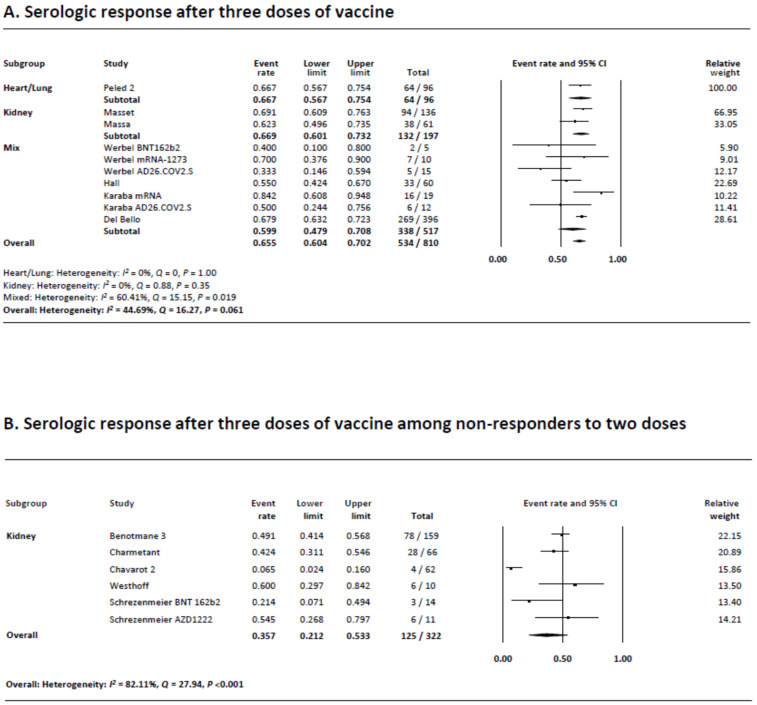
(**A**) Meta-analysis of serological response after three doses of vaccine. (**B**) Meta-analysis of serological response after three doses of vaccine among non-responders to two doses.

**Table 1 viruses-14-01822-t001:** Characteristics and outcomes of the included studies.

Author	Country	Year	Patient Numbers and Description	Control Numbers and Description		Age of Patients (years)		Sex of Patients (% Females)	Cases, % of Patients on Immunosuppression		Cases, % of Patients on Steroids	Type of Vaccine	Number of Patients Receiving 1 Dose	Number of Patients Receiving 2 Doses	Number of Patients Receiving 3 Doses
1 **Grupper** **(Full paper)**	Israel	2021	136 (Kidney 100%)	25 (HCWs)		Cases mean: 58.6 (SD 12.7), Controls mean: 52.7 (SD 11.5)		Cases 18.3%, Controls 68%	ATG last 12 months 7.35%, Rituximab last 12 months 2.9%, CNIs 90.4%, mTORs 7.35%, MMF 76.5% Triple maintenance immunosuppression 78.8%		High-dose steroids last 12 months 23.5%, Low-dose prednisone 89.0%	BNT162b2 (Pfizer-BioNTech) 100%	NA	161	NA
2 **Boyarsky 1** **(Letter)**	United States	2021	436 (Kidney 50.2%, Liver 17.9%, Heart 15.1%, Lung 11.2% Pancreas 1.1%, Multiorgan 3.2%)	None		Median: 55.9 (IQR 41.3–67.4)		61%	100% (Tacrolimus 83%, MMF 66%, Azathioprine 9%, Sirolimus 4%, Everolimus 2%)		54%	BNT162b2 (Pfizer-BioNTech) 52%, mRNA-1273 (Moderna) 48%	436	NA	NA
3. **Wadei** **(Letter)**	United States	2021	7 (Double lung 14.3%, Kidney 28.6%, Heart and kidney 14.3%, Kidney and pancreas 14.3%)	None		Mean: 59 (Range 42–69)		0%	100% (Tacrolimus 85.7%, MMF 100%, Belatacept 14.3%)		100%	BNT162b2 (Pfizer-BioNTech) 57.1%, mRNA-1273 (Moderna) 42.9%	2	5	NA
4. **Rincon-Arevalo** **(Full paper)**	Germany	2021	40 (Kidney 100%)	35 (Mainly HCWs)		Cases median: 62.4 (IQR 51.3–69.5), Controls median: 51.0 (IQR 34.0–80)		Cases 30%, Controls 42.9%	100% (MMF 97.5%, Tacrolimus 55%, Cyclosporine 37.5%, Azathioprine 0.9%, mTOR inhibitors 3.7%)		92.5%	BNT162b2 (Pfizer-BioNTech) 100%	NA	75	NA
5. **Benotmane 1** **(Letter)**	France	2021	242 (Kidney 100%)	None		57.7 (49.3–67.6)		35.50%	100% (Induction treatment; ATG 59.5%, anti-CD25 37.9%, no induction 2.6%, CNIs; Tacrolimus 55.2%, cyclosporine 34%, no CNI 10.8%, Others; MMF/MPA 79.3%, Azathioprine 2.9%, mTOR inhibitors 14.5%, Belatacept 3.8%)		58.9%	mRNA-1273 (Moderna) 100%	242	NA	NA
6. **Benotmane 2** **(Letter)**	France	2021	205 (Kidney 100%)	None		57.7 (49.4–67.5)		36.60%	100% (Induction treatment: ATG 60.5%, anti-CD25 35.9%, no induction 3.6%, CNIs; Tacrolimus 56.4%, Cyclosporine 35.8%, no CNI 7.8%, Others; MMF/MPA 78.9%, Azathioprine 2.9%, mTOR inhibitors 13.2%, Tacrolimus + MMF/MPA 48%, Tacrolimus + MMF/MPA + steroids 31.3%, Belatacept 2.5%)		59.8%	mRNA-1273 (Moderna) 100%	NA	205	NA
7. **Rabinowich** **(Full paper)**	Israel	2021	80 (Liver 100%)	25 (HCWs)		Cases mean: 60.1 (SD 12.8) Controls mean: 52.7 (SD 11.5)		Cases 30%, Controls 68%	97.5% (Tacrolimus 81.3%, Cyclosporine 12.5%, Everolimus 22.5%, Azathioprine 5%, MMF 50%)		High dose steroids last 12 months 20%, prednisone 30%	BNT162b2 (Pfizer-BioNTech) 100%	NA	105	NA
8. **Yi** **(Letter)**	United States	2021	145 (Kidney 100%)	31 (ESRD patients, 4 on immunosuppression)		NA		NA	100%		NA	BNT162b2 (Pfizer-BioNTech), mRNA-1273 (Moderna)	176	NA	NA
9. **Peled** **(Full paper)**	Israel	2021	77 (Heart 100%)	136 (Healthy controls)		Cases median: 62.0 (49.0–68.0) Controls mean: 63 (SD 13)		Cases 36%, Controls 63%	100% (MPA 53.2%, MMF 22.1%, Everolimus 26.0%)		75.3%	BNT162b2 (Pfizer-BioNTech) 100%	NA	213	NA
10. **Sattler** **(Full paper)**	Germany	2021	39 (Kidney 100%)	39 (HCWs)		Cases mean: 57.38 (SD 14.04) Controls mean: 53.03 (SD 17.58)		Cases 28.21%, Controls 48.72%	100%		89.7–97.4%	BNT162b2 (Pfizer-BioNTech) 100%	NA	78	NA
11. **Marion** **(Letter)**	France	2021	367 (Kidney 73.8%, Liver 15.8%, Thoracic organs 9.0%, Pancreas 1.4%) (Includes 5 patients with prior COVID-19 exposure)	None		Cases mean: 59 (Standard error 1)		36.8%	100% (Tacrolimus 78.2%, Cyclosporine 7.1%, MPA 68.4%, mTOR inhibitors 25.6%, Belatacept 9.3%)		81.7%	mRNA vaccine 100%: Of original cohort of 950, 942 received BNT162b2 (Pfizer-BioNTech) and 8 received mRNA-1273 (Moderna)	367	367	NA
12. **Miele** **(Letter)**	Italy	2021	16 (Kidney 31.3%, Lung 31.3%, Liver 25%, Heart 12.5%)	23 (HCWs)		Cases mean: 57 (SD 15.9) Controls mean: 44 (SD 7.2)		Cases 18.8%, Controls 56.5%	Tacrolimus 93.7%, Everolimus 6.3%, MMF 62.5%		56.3%	BNT162b2 (Pfizer-BioNTech) 100%	NA	39	NA
13. **Havlin** **(Communication)**	Czech Republic	2021	48 (Lung 100%)	10 (Healthy volunteers)		Cases mean: 52.1 (SD 14.3) Controls median: 39.8 (IQR 33.3–47.8)		Cases 39.6%, Controls NA	100% (Tacrolimus 97.9%, Cyclosporine 2.1%, MMF 91.7%)		97.9%	BNT162b2 (Pfizer-BioNTech) 100%	46	30	NA
14. **Rosen-Zvi** **(Full paper)**	Israel	2021	308 (Kidney 100%)	None		Mean: 57.51 ± 13.84		36%	100% (MPA 73.4%, Tacrolimus 92.5%, Cyclosporine 7.5%, mTOR inhibitor 8.4%, Rituximab 1.9%, ATG 4.5%)		8.4%	BNT162b2 (Pfizer-BioNTech) 100%	NA	308	NA
15. **Shostak** **(Letter)**	Israel	2021	168 (Lung 100%)	None		Median: 60.5 (IQR 49.3–67.8)		33%	Includes mTOR inhibitors (patients treated with combination therapy of CNI and Everolimus) 17%, includes antimetabolite (patients treated with MMF/MPA or Azathioprine) 92%		Mean prednisone dose (5.0 mg, IQR 5.0–10.0)	BNT162b2 (Pfizer-BioNTech) 100%	168	168	NA
16. **Bertrand** **(Full paper)**	France	2021	45 (Kidney 100%)	None		Mean: 63.5 ± 16.3		48.9%	100% (Tacrolimus 53.3%, Cyclosporine 17.8%, MMF 82.2%, Azathioprine 8.9%, Everolimus 6.7%, Belatacept 22.2%)		46.7%	BNT162b2 (Pfizer-BioNTech) 100%	45	45	NA
17. **Cucchiari** **(Full paper)**	Spain	2021	117 (Kidney 93.2%, Kidney and pancreas 6.8%)	None		Mean: 59.00 ± 52.42		32.3%	100% (Tacrolimus 83.8%, Cyclosporine 4.3%, MMF 61.5%, mTOR inhibitors 32.5%, Azathioprine 3.4%, Belatacept 6.8%, Eculizumab 1.7%)		79.5%	mRNA-1273 (Moderna) 100%	NA	148	NA
18. **Husain** **(Letter)**	United States	2021	28 (Kidney 100%) (Includes 3 patients with prior COVID-19 exposure)	None		Median: 66 (Range 42–87)		39%	Tacrolimus 75%, Belatacept 21%, MMF/MPA 61%, Azathioprine 11%, Leflunomide 4%, Sirolimus/everolimus 14%		32%	BNT162b2 (Pfizer-BioNTech) 57%, mRNA-1273 (Moderna) 43%	NA	28	NA
19. **Korth** **(Communication)**	Germany	2021	23 (Kidney 100%)	23 (HCWs)		Cases mean: 57.7 ± 13.5 Controls mean: 44.4 ± 9.2		Cases 52%, Controls 61%	MMF 78.3%, Tacrolimus 60.9%, Cyclosporine 17.4%, Sirolimus 21.7%, Everolimus 4.3%, Belatacept 4.3%, Azathioprine 4.3%		60.8%	BNT162b2 (Pfizer-BioNTech) 100%	NA	46	NA
20. **Boyarsky 2** **(Letter)**	United States	2021	658 (Kidney 48.9%, Liver 19.6%, Heart 14.7%, Lung 10.8%, Pancreas 0.8%, Multiorgan 4.0%)	None		Median: ≥60 (Range 18—≥60)		58.7%	100% (Antimetabolites 71.9%, Other 28.1%)		NA	BNT162b2 (Pfizer-BioNTech) 52.0%, mRNA-1273 (Moderna) 46.7%	658	658	NA
21. **Mazzola** **(Full paper)**	France	2021	143 (Liver 40.6%, Kidney 41.3%, Heart 18.2%) (Includes 8 patients with prior COVID-19 exposure)	25 (HCWs)		Cases median: 61.0 (IQR 55.0–67.0) Controls median: 55.0 (IQR 38.0–62.0)		Cases 28.7%, Controls 72%	CNIs 82.5%, MMF 72.0%, mTOR inhibitor 18.9%, Tri-therapy 50.4%		62.2%	BNT162b2 (Pfizer-BioNTech) 100%	125	158	NA
22. **Firket** **(Letter)**	Belgium	2021	10 (Kidney 100%)	10 (Belgian vaccination program)		Cases mean: 49.7 (SD 13.8) Controls mean: 51.5 (SD 10.5)		Cases 50%, Controls 30%	100% (CNIs 100%, Antimetabolites 100%)		40%	BNT162b2 (Pfizer-BioNTech) 100%	20	20	NA
23. **Danthu (Communication)**	France	2021	74 (Kidney 100%)	7 (HCWs)		Cases mean: 64.8 ± 11.5 Controls mean: 51.6 ± 6.8		Cases 40.5%, Controls 42.9%	100% (CNIs 91.8%, Belatacept 2.7%, Everolimus 10.8%, MMF 70.3%, MPA 9.5%, Azathioprine 2.7%)		45.9%	BNT162b2 (Pfizer-BioNTech) 100%	NA	81	NA
24. **Boyarsky 3** **(Letter)**	United States	2021	12 (Kidney 58%, Liver 25%, Heart 8%, Lung 8%)	None (compared to mRNA cohort data)		Median: 56 (IQR 42–60)		58.3%	100% (Rapamycin 8.3%, Azathioprine 16.7%, Tacrolimus 83.3%, MMF 66.7%, Everolimus 8.3%)		58.3%	Ad26.COV2.S (Janssen/Johnson & Johnson) 100%	12	NA	NA
25. **Marinaki** **(Letter)**	Greece	2021	34 (Heart 70.6%, Kidney 29.4%)	116 (HCWs)		Cases median: 60 (IQR 49.1–68.4) Controls: Age and sex matched HCW		Cases 20.6%, Controls Age and sex matched HCW	100% (CNIs 94%, Antimetabolite therapy 44%, mTOR inhibitor 62%)		15%	BNT162b2 (Pfizer-BioNTech) 100%	NA	150	NA
26. **Chavarot 1** **(Letter)**	France	2021	101 (Kidney 100%)	None		Cases median: 64 (53–73)		32.7%	100% (Belatacept 100%, MPA 78.2%, mTOR inhibitors 11.9%, Tacrolimus 7.9%, Azathioprine 2.0%)		96.0%	BNT162b2 (Pfizer-BioNTech) 100%	101	35	NA
27. **Ou** **(Full paper)**	United States	2021	609 (Kidney 100%, Pancreas 6%, Liver 4%, Heart 2%, Lung 1%)	None		Median: 58 (IQR 45–68)		59.2%	100% (Belatacept 3.9%, MMF 71.9%, Tacrolimus 77.2%, Azathioprine 9,7%, Sirolimus 8.4%)		68.5%	BNT162b2 (Pfizer-BioNTech) 51.9%, mRNA-1273 (Moderna) 44.8%	592	400	NA
28. **Kamar** **(Correspondence)**	France	2021	101 (Kidney 77.2%, Liver 11.9%, Lung 7.9%, Pancreas 3.0%)	None		Mean: 58 ± 2		30.3%	CNIs 79%, Anti-metabolites 66%, mTOR inhibitors 30%, Belatacept 12%		87%	BNT162b2 (Pfizer-BioNTech) 100%	101	99	99 (Included in Del Bello)
29. **Hallett** **(Full paper)**	United States	2021	237 (Heart 57%, Lung 43%)	None		Median: 62 (46–69) (Heart 60 (44–69), Lung 63 (48–70))		55% (Heart 51%, Lung 59%)	100% (Tacrolimus 86%, MMF 62%, Sirolimus 14%, Cyclosporine 8%, Azathioprine8%, Everolimus7%, Belatacept 1%)		57%	BNT162b2 (Pfizer-BioNTech) 53%, mRNA-1273 (Moderna) 47%	237	237	NA
30. **Stumpf** **(Full paper)**	Germany	2021	368 (Kidney 100%)	144 (HCWs)		Cases mean: 57.3 ± 13.7 Controls mean: 48 ± 11.9		34.5%	99.7% (CNIs 87.5%, MMF 76.1%, mTOR Inhibitor 16%, Belatacept 4.6%)		48.4%	BNT162b2 (Pfizer-BioNTech) 28%, mRNA-1273 (Moderna) 72%	144	333	NA
31. **Zadok** **(Short report)**	Israel	2021	42 (Heart 100%)	None		Median: 61 (IQR 44–69)		17%	99.7% (CNIs 81%, MMF 55%%, mTOR Inhibitor 57%,)		69%	BNT162b2 (Pfizer-BioNTech) 100%	42	NA	NA
32. **Schramm** **(Full paper)**	Germany	2021	50 (Heart 84%, Lung 14%, Heart/lung 2%)	50 (HCWs)		Mean: 55 ± 10		36%	100% (Tacrolimus/MMF 82%, Cyclosporine/MMF 10%, Tacrolimus/mTOR-Inhibitor 8%)		NA	BNT162b2 (Pfizer-BioNTech) 100%			
33. **Werbel** **(Letter)**	United States	2021	30 (Kidney 73.3%, Heart 6,7%, Lung 3.3%, Liver 10%, Pancreas 3.3%, Kidney and pancreas 3.3%)	None		Median: 57 (IQR 44–62)		56.7%	Tacrolimus or Cyclosporine + MMF 83.3%, Sirolimus 3.3%, Belatacept 3.3%		80%	Initial doses: BNT162b2 (Pfizer-BioNTech) 56.7%, mRNA-1273 (Moderna) 43.3% Third dose: BNT162b2 (Pfizer-BioNTech) 16.7%, mRNA-1273 (Moderna) 33.3%, AD26.COV2.S (Janssen/Johnson & Johnson) 50% Combinations: Pfizer-BioNTech + Janssen/Johnson & Johnson 23.3%, Pfizer-BioNTech + Moderna 23.3%, Pfizer-BioNTech + Pfizer-BioNTech 10%, Moderna + Janssen/Johnson & Johnson 26.7%, Moderna + Moderna 10%, Moderna + Pfizer-BioNTech 6.7%	NA	30	30
34. **Hall** **(Correspondence)**	Canada	2021	120 (Lung 24.2%, Heart 15%, Kidney 24.2%, Liver 16.7%, Pancreas/kidney-pancreas 20%) (60 experimental, 60 placebo)	None (Randomized controlled trial contained all solid organ transplant patients)		Third dose median: 66.9 (IQR 64.0–71.8), Placebo median: 65.9 (IQR 62.9–70.3)		Third dose: 38.3%, Placebo: 30%	100% (Tacrolimus 77.5%, Cyclosporine 20.8%, Sirolimus 9.2%, MMF 75%, Azathioprine 10%)		Third dose 83.3%, Placebo 70%	mRNA-1273 (Moderna) 100%	NA	120	60
35. **Karaba** **(Preprint)**	United States	2021	31 (Kidney 61.3%, Liver 22.6%, Heart 9.7%, Pancreas 3.2%, Lung 3.2%)	None (15 Healthy controls receiving 2 mRNA vaccine doses)		Median: 60 (IQR 49–67)		Cases 54.8%, Controls 33.3%	(CNI 80.6%, mTORi 9.7%, Anti-metabolites 64.5%)		51.6%	Transplant recipients: First two doses: mRNA vaccine 100%, Third dose: mRNA vaccine (19, 61.3%), Ad26.COV2.S (Janssen/Johnson & Johnson) (12, 38.7%) Healthy controls: 2 doses of an mRNA vaccine (15, 100%)	NA	46	46
36. **Peled 2** **(Full paper)**	Israel	2021	96 (Heart 100%)	None		Median: 61.0 (IQR 49.8–68.0)		29.2%	100% (Tacrolimus 82.3%, Mycophenolate sodium 54.2%, MMF 24.0%, Cyclosporine 11.5%, Everolimus 21.9%)		80.2%	BNT162b2 (Pfizer-BioNTech) 100%	NA	96	96
37. **Benotmane 3** **(Letter)**	France	2021	159 (Kidney 100%)	None		Median: 57.6 (IQR 49.6–66.1)		38.4%	100% (Tacrolimus + MMF/MPA + steroids 52.8%, All other regimens 47.2%)		Tacrolimus + MMF/MPA + steroids (84, 52.8%) All other regimens (75, 47.2%)	mRNA-1273 (Moderna) 100%	NA	159	159
38. **Masset** **(Letter)**	France	2021	136 (Kidney 91.2%, kidney-pancreas/pancreas 8.8%)	None		Mean: 63.7 (SD 11.7)		36.8%	(CNI 84.6%, mTORi 14.7%, Antimetabolites 74.3%, NA 1.47%)		31.6% (NA 1.47%)	BNT162b2 (Pfizer-BioNTech) 100%	NA	456	136
39. **Del Bello** **(Letter)**	France	2021	396 (Kidney 69.9%, Liver 17.4%, Heart 8.33%, Lung 0.3%, Pancreas 1.5%, Multiple organs 2.5%)	None		Mean: 59 (SD 15)		34.8%	100% (CNI 86.1%, MPA 72.0%, mTORi 26.8%, Belatacept 8.8%)		82.1%	BNT162b2 (Pfizer-BioNTech) 100%	NA	396	396
40. **Charmetant** **(Preprint)**	France	2021	66 (Kidney 100%)	None		Mean: 56.3 (SD 12.3)		56.1%	100% (CNI 92.4%, MMF/MPA 81.8%, mTORi 7.6%, Belatacept 1.5%)		86.4%	BNT162b2 (Pfizer-BioNTech) 100%	NA	66	66
41. **Chavarot 2** **(Communication)**	France	2021	62 (Kidney 100%)	None		Median 63.5 years (IQR 51–72)		41.9%	100% (Belatacept 100%, Everolimus 12.9%, MPA 71.0%, Azathioprine 4.8%, CNI 3.2%)		100%	BNT162b2 (Pfizer-BioNTech) 100%	NA	62	62
42. **Westhoff** **(Letter)**	Germany	2021	10 (Kidney 100%)	None		Mean: 59.5 (Range 41–76)		20%	100% (CNI 80%, mTORi 10%, Belatacept 10%, MPA 90%)		100%	First two doses: BNT162b2 (Pfizer-BioNTech) 100% Third dose: mRNA-1273 (Moderna) 100%	10	10	10
43. **Massa** **(Preprint)**	France	2021	61 (Kidney 100%)	None		Median 58 years (IQR 47.1–66.1)		27.9%	100% (Antimetabolites 62.3%, CNI 93.4%, mTORi 9.8%, Belatacept 1.6%)		88.5%	BNT162b2 (Pfizer-BioNTech) 100%	NA	61	61
44. **Schrezenmeier** **(Preprint)**	Germany	2021	25 (Kidney 100%)	None		Mean: 59.7 (SD 13.8)		44.0%	100% (Tacrolimus 56%, Cyclosporine 32%, MMF 96%, mTORi 16%)		96%	First two doses: BNT162b2 (Pfizer-BioNTech) 100% Third dose: BNT162b2 (Pfizer-BioNTech) 56%, AZD1222 (Oxford-AstraZenaca) 44%	25	25	25
**Author**	**Used to Check Antibody Response**		**Timing of Test**	**After One Dose**				**After Two Doses**				**After Three Doses**			
				**Cases Responders**	**Controls Responders**	**Cases Ab Titers**	**Controls Ab Titers**	**Cases Responders**	**Controls Responders**	**Cases Ab Titers**	**Controls Ab Titers**	**Cases Responders**	**Controls Responders**	**Cases Ab Titers**	**Controls Ab Titers**
1. **Grupper**	DiaSorin LIAISON SARS-CoV-2 S1/S2 IgG chemiluminescent assay		10–20 days after the second dose	-	-	-	-	51/136	25/25	Median: 71.8 AU/mL (IQR 37.6–111.7 AU/mL)	Median: 189.0 AU/mL (IQR 141.10–248 AU/mL)	-	-	-	-
2. **Boyarsky 1**	EUROIMMUN anti-S1 IgG assay or Roche Elecsys anti-RBD pan-Ig assay		14–21 days after the first dose	76/436	-	NA	-	-	-	-	-	-	-	-	-
3. **Wadei**	Antispike antibody, manufacturer NA		Median 28 days (Range 6–44 days) after the first dose	0/2	-	NA	-	1/5	-	1.4 U/mL	-	-	-	-	-
4. **Rincon-Arevalo**	EUROIMMUN anti-SARS-CoV-2 ELISA, GenScript Surrogate SARS-CoV-2 virus neutralization ELISA		7 ± 2 days after the second dose				-	4/40 (IgG 1/40, IgA 4/40, Neutralizing antibodies 0/40)	35/35	IgG median: 0.09 (IQR 0.07–0.15) IgA median: 0.20 (IQR 0.15–0.40) NC median: 0.07 (IQR 0.05–0.13)	IgG median: 58.59 (IQR 31.90–71.96) IgA median: 41.10 (IQR 27.03–58.37) NC median: 0.08 (IQR 0.06–0.11)	-	-	-	-
5. **Benotmane 1**	ARCHITECT IgG II Quant test		28 days after the first dose	26/242	-	Median: 224 AU/mL (IQR 76–496 AU/mL)	-	-	-	-	-	-	-	-	-
6. **Benotmane 2**	Abbott ARCHITECT IgG II Quant test		1 month after the second dose	-	-	-	-	98/205	-	Median: 803.2 AU/mL (IQR 142.6–4609.6 AU/mL)	-	-	-	-	-
7. **Rabinowich**	DiaSorin LIAISON SARS-CoV-2 S1/S2 IgG chemiluminescent assay		Cases: Mean 14.8 (±3.2) days after the second dose, Controls: Mean 15.8 (±2.9) days after the second dose				Median: 1:150 COVID-19 IgG titer (Range 1:50 to >1:1350)	38/80	25/25	Mean: 95.41 (±92.4) AU/mL	Mean: 200.5 (±65.1) AU/mL	-	-	-	-
8. **Yi**	Anti-SARS-CoV-2 IgG and total antibody, anti-SARS-CoV-2 Nucleocapsid IgG, and anti-Spike IgG titer		At the time of second dose	9/145	27/31	Median: 1:150 COVID-19 IgG titer (Range 1:50 to <1:450)	Median: 1:150 COVID-19 IgG titer (Range 1:50 to >1:1350)	-	-	-	-	-	-	-	-
9. **Peled 1**	An "in-house" ELISA that detects IgG antibodies against SARS-CoV-2 RBD. A SARS-CoV-2 pseudo-virus (psSARS-2) neutralization assay was performed to detect SARS-CoV-2 neutralizing antibodies using a propagation-competent VSV-spike		Transplant: 21 ± 10 days after the second dose Control: 13.3 ± 1.4 days after the second dose	-	-	-	-	14/77	134/136	NA	NA	-	-	-	-
10. **Sattler**	Euroimmun ELISA-based analysis of SARS-CoV2 spike S1 domain-specific IgG and IgA		8 ± 1 days after the second dose	-	-	-	-	4/39 (IgG 1/39, IgA 4/39, Neutralizing antibodies 0/39)	39/39	NA	NA	-	-	-	-
11. **Marion**	Beijing Wantai Biological Pharmacy Enterprise SARS-CoV-2 total antibodies ELISA or another validated anti–SARS-CoV-2 spike protein assay		28 days after the first or second dose	23/367	-	NA	-	124/367	-	NA	-	-	-	-	-
12. **Miele**	DiaSorin LIAISON SARS-CoV-2 S1/S2-IgG chemiluminescent assay		Cases: Median 20 days (Range 15–76) after the second dose, Controls: Median 15 days (Range 15–20) after the second dose	-	-	-	-	6/16	23/23	Median: 3.8 AU/mL Mean: 87.32 AU/mL	Median: 212 AU/mL Mean: 233 AU/mL	-	-	-	-
13. **Havlin**	Euroimmun anti-SARS-CoV-2 Spike S1 IgG ELISA, confirmed independently by TestLine Microblot-Array COVID-19 IgG and DiaSorin Liaison SARS-CoV-2 Trimeric S IgG		Cases: At baseline, before the second dose, 7 days after the second dose, 4–6 weeks after the second dose, Controls: 31 days (IQR 19–41 days) after the second dose	0/46	-	NA	-	7 days after second vaccination: 0/30 (4–6 weeks after second vaccination: 0/21)	10/10	NA	NA	-	-	-	-
14. **Rosen-Zvi**	Abbott SARS-CoV-2 IgG II Quant assay		Median 28 days (IQR 22–34 days) after the second dose	-	-	-	-	112/308	-	Median: 15.5 AU/mL (IQR 3.5–163.6 AU/mL)	-	-	-	-	-
15. **Shostak**	Abbott SARS-CoV-2 IgG II Quant assay		Median 16 days (IQR 15–18) after the second dose	6/168	-	Geometric mean S-IgG titer: 3.12 (SD 4.05)	-	31/168	-	Geometric mean S-IgG titer: 9.29 (SD 9.22)	-	-	-	-	-
16. **Bertrand**	Abbott ARCHITECT IgG II Quant test		Three weeks after the first dose and one month after the second dose	1/45	-	311 AU/mL	-	8/45	-	Responder median: 671 AU/mL (IQR 172–1523 AU/mL)	-	-	-	-	-
17. **Cucchiari**	A serological assay based on the Luminex technique measuring antibodies against the Receptor-Binding Domain (RBD) of the spike glycoprotein of SARS-CoV-2		2 weeks after the second dose	-	-	-	-	35/117 (IgG/IgM 5/117, IgG 27/117, IgM 3/117)	-	NA	-	-	-	-	-
18. **Husain**	DiaSorin LIAISON anti-S IgG immunoassay or Roche Diagnostics Elecsys anti-S IgG immunoassay		Median 29 days (Range 12–59) after the second dose	-	-	-	-	7/28	-	NA	-	-	-	-	-
19. **Korth**	DiaSorin LIAISON® SARS-CoV-2 TrimericS IgG assay		Cases: Mean 15.8 ± 3.0 days after the second dose Controls: Mean 13.7 ± 1.8 days after the second dose	-	-	-	-	5/23	23/23	Mean: 50.9 ± 138.7 AU/mL	Mean: 727.7 ± 151.3 AU/mL	-	-	-	-
20. **Boyarsky 2**	EUROIMMUN anti-S1 IgG assay 28.6%, Roche Elecsys anti-RBD pan-Ig assay 71.4%		Median 21 days (IQR 18–25) after the first dose, Median 29 days (IQR 28–31) after the second dose	98/658	-	NA	-	357/658	-	Roche median: 2.14 U/mL (IQR <0.4–245.8), EUROIMMUN median: 1.23 AU (IQR 0.13–6.38)	-	-	-	-	-
21. **Mazzola**	Abbott Diagnostics Alinity I chemiluminescent microparticle immunoassays		28 days after the first and second dose	9/125	-	Responder median: 153 AU/mL (IQR 129–860 AU/mL)	-	38/133	25/25	Responder median: 759 AU/mL (IQR 257–3269 AU/mL)	NA	-	-	-	-
22. **Firket**	DiaSorin LIAISON® chemiluminescence immunoassay		At time of second dose, ~15 days after the second dose, ~50 days after the second dose for transplant patients	0/10	9/10	Median: 0 AU/mL (Range 0–0 AU/mL)	Median: 35.5 AU/mL (Range 0–118 AU/mL)	15 days after second dose 1/10 (50 days after second dose 3/10)	15 days after second dose: 10/10	15 days after second dose median: 0 AU/mL (0–60 AU/mL) 50 days after second dose median: 0 AU/mL (0–46 AU/mL)	15 days after second dose median: 263 AU/mL (Range 153–2090 AU/mL)	-	-	-	-
23. **Danthu**	DiaSorin LIAISON SARS-CoV-2 TrimericS IgG		14, 28, 36, and 58 days after the first dose	-	-	-	-	3/74	7/7	NA (low number of responders)	Day 14: 59 AU/mL (IQR 26.5–216.5 AU/mL) Day 36: 1082 AU/mL (IQR 735.0–1662 AU/mL) Day 58: 925 AU/mL (IQR 637–3624.5 AU/mL)	-	-	-	-
24. **Boyarsky 3**	Roche Elecsys anti-SARS-CoV-2 S enzyme immunoassay		Median 33 days (IQR 31–44 days) after the first dose	2/12	-	Median: 2.39 U/mL (Range 1.33–3.45 U/mL)	-	-	-	-	-	-	-	-	-
25. **Marinaki**	Abbott SARS-CoV-2 IgG II Quant anti-SARS-CoV-2-RBD IgG assay		Median 10 days (IQR 9–10 days) after thesecond dose	-	-	-	-	20/34	116/116	Median: 1370 AU/mL Geometric mean: 948 AU/mL	Median: 11,710 AU/mL Geometric mean: 11,300 AU/ML	-	-	-	-
26. **Chavarot**	Abbott SARS-CoV-2 IgG II Quant antibody test or Beijing Wantai Biological Pharmacy Enterprise SARS-CoV-2 total antibodies ELISA		28 and 60 days after first dose	2/101	-	NA	-	2/35	-	NA	-	-	-	-	-
27. **Ou**	EUROIMMUN anti-S1 IgG assay or Roche Elecsys anti-RBD pan-Ig assay		After 1 dose: Non-belatacept: Median 21 days (IQR 19–26) Belatacept: Median 22 days (IQR 19 -26) After 2 doses: Non-belatacept: Median 29 days (IQR 28–32) Belatacept: Median 29 days (IQR 28–31)	77/592 (Belatacept 0/24, Non-belatacept 77/568)	-	EUROIMMUN median IgG titer: 2.33 AU (IQR 1.68 – 4.77) Roche median IgG titer: 4.24 U/mL (IQR 1.81 – 15.05)	-	191/400 (Belatacept 1/19, Non-belatacept 190/381)	-	Non-belatacept: EUROIMMUN median IgG titer: 6.23 AU (IQR 3.12 – 8.74 AU) Roche median IgG titer: 78.10 U/mL (IQR 7.42 – 250 U/mL) Belataceot: 48.07 U/mL	-	-	-	-	-
28. **Kamar**	Beijing Wantai Biological Pharmacy Enterprise SARS-CoV-2 total antibodies ELISA		4 weeks after the third dose	4/101	-	NA	-	40/99	-	NA	-	67/99 (Included in Del Bello)			
29. **Hallett**	Roche Elecsys for antibodies against the receptor-binding domain (RBD) or EUROIMMUN for antibodies to the S1 domain		Median of 21 days (IQR 19–26 days) after the first dose, median of 29 days (IQR 28–32 days) after the second dose	28/237 (Heart 19/134, Lung 9/103)	-	-	-	120/237 (Heart 83/134, Lung 37/103)	-	Anti-spike RBD assay: 250 U/mL (IQR, 174–250 U/mL) for first dose responders, 23.8 U/mL (IQR, 3.9–244.2 U/mL) for second dose responders, and 0 U/mL (IQR, 0–0 U/mL) for non-responders	-	-	-	-	-
30. **Stumpf**	SARS-CoV-2 specific IgG- or IgA-antibody reactions (Euroimmun) against the Spike protein subunit S1		3–4 weeks after the first dose, 4–5 weeks after the second dose	11/144	53/55	NA	NA	140/333	132/134	NA	NA	-	-	-	-
31. **Zadok**	Anti-spike IgG (S-IgG) antibodies		21–26 days and 35–40 days after the first dose	6/39	-	NA	-	18/37	-	NA	-	-	-	-	-
32. **Schramm**	SARS-CoV-2 IgG II Quant assay (Abbott) which was used for the quantitative measurement of IgG antibodies against the spike receptor-binding domain (RBD)		21 days after the first and the second dose	2/50	49/50	NA	Median 82 (41;149) BAU/ml	5/50	50/50	NA	median 1417 (732; 2589) BAU/ml	-	-	-	-
33. **Werbel**	EUROIMMUN anti-S1 IgG assay or Roche Elecsys anti-RBD pan-Ig assay		Median 9 days (IQR 2–33) before the third dose, Median 14 days (IQR 14–17 days) after the third dose	-	-	-	-	6/30	-	EUROIMMUN median 0.15 AU, Roche median 0.4 U/mL	-	14/30 (BNT162b2 2/5, mRNA-1273 7/10, Ad26.COV2.S: 5/15)	-	EUROIMMUN median 0.37 AU, Roche median NA	-
34. **Hall**	Roche Elecsys anti-SARS-CoV-2 S enzyme immunoassay		1 month after third vaccination	-	-	-	-	12/120 (Third dose: 7/60 Placebo third dose: 5/60)	-	Third dose: Median 0.37 U/mL (IQR 0.2 – 27.64) Placebo: Median 0.44 U/mL (IQR 0.2 – 18.19)	-	Third dose: 33/60 (Placebo third dose: 10/57)	-	Third dose: Mean: 3145 U/mL (SD 7517), Median: 313.8 U/ML (IQR 0.2–2191)Placebo: Mean: 86 U/mL (SD 231), Median: 1.19 U/mL (IQR 0.2–63.4)	-
35. **Karaba**	Meso Scale Diagnostics (MSD) V-PLEX COVID-19 155 Respiratory Panel 3 multiplex chemiluminescent kit		14 days after third vaccination	-	-	-	-	12/31 (Anti-RBD IgG: 12/31, Anti-S IgG: 8/31, Anti-N IgG: 0/31)	15/15	NA	NA	24/31 (Anti-RBD IgG: 24/31, Anti-S IgG: 22/31 (mRNA: 16/19 Ad26.COV2.S: 6/12))	-	NA	-
36. **Peled 2**	An "in-house" enzyme-linked immunosorbent assay that detects IgG antibodies against SARS-CoV-2 RBD. A SARS-CoV-2 pseudo-virus (psSARS-2) neutralization assay was performed to detect SARS-CoV-2 neutralizing antibodies using a propagation-competent VSV-spike		Mean 17.5 days (SD 3.9) after third vaccination	-	-	-	-	26/96 (May overlap with previous Peled 1 study)	-	IgG GMT: 0.49 (95% CI 0.39–0.62) Neutralizing antibody GMT: 3.05 (95% CI, 2.05–4.55)	-	64/96	-	IgG GMT: 1.58 (95% CI 1.24–2.00)Neutralizing antibody GMT: 27.25 (95% CI, 15.70–47.30)	-
37. **Benotmane**	Abbott ARCHITECT IgG II Quant test		Median 28 days (IQR 27–33) after third vaccination	-	-	-	-	0/159 (95 pts had no antibody response, 64 pts had an antibody response below the seropositivity threshold of 50 AU/mL)	-	<50 AU/mL	-	78/159	-	Responder median: 586 AU/mL (IQR 197.2–1920.1)Non-responder median: <50 AU/mL	-
38. **Masset**	Abbott Architect chemiluminescent microparticle immunoassay, Siemens Atellica chemiluminescence immunoassay, Roche Elecsys electrochemiluminescence immunoassay		Median 30 days (IQR 28–32)	-	-	-	-	227/456 (Assessed after second and third dose: 34/85)	-	NA	-	94/136	-	NA	-
39. **Del Bello**	Beijing Wantai enzyme-linked immunosorbent assay (228, 57.6%) or other anti-SARS-CoV-2 spike assay (168, 42.4%)		4 weeks after third vaccination	-	-	-	-	164/396	-	NA	-	269/396	-	NA	-
40. **Charmetant**	Snibe Diagnostic Maglumi SARS-CoV-2 S-RBD IgG test on a Maglumi 2000 analyser		14 days after third vaccination	-	-	-	-	27/93	-	<142 BAU/mL	-	28/66 (All 66 had no response after two doses)	-	NA	-
41. **Chavarot 2**	Abbott SARS-CoV-2 IgG II Quant antibody test		Median 28 (IQR 28–33) days after third vaccination	-	-	-	-	0/62	-	<50 AU/mL	-	4/62	-	Median 209 AU/mL (IQR 20–409 AU/ml)	-
42. **Westhoff**	Roche Elecsys anti-SARS-CoV-2 S enzyme immunoassay		14 days after third vaccination	0/10	-	<0.8 U/mL	-	0/10	-	<0.8 U/mL	-	6/10	-	Responder median 542 U/mL (IQR 478–923)	-
43. **Massa**	Abbott ELISA on the Abbott Architect I1000 analyser		28 days after third vaccination	-	-	-	-	27/61	-	GMT IgG: 528.3 AU/mL (95% CI 300.0–930.1)	-	38/61	-	GMT IgG: 2395 AU/mL (95% CI 1214–4724)	-
44. **Schrezenmeier**	Euroimmun ELISA-based analysis of SARS-CoV2 spike S1 domain-specific IgG and IgA		7 ± 2 days after the second and third vaccination, and 19–27 days after each vaccination	0/25	-	NA	-	0/25	-	NA	-	9/25 (BNT162b2: 3/14, AZD1222: 6/11)	-	NA	-

HCW, health care worker; ATG, anti-thymocyte globulin; MMF, mycophenolate mofetil; MPA, mycophenolic acid; mTOR, mechanistic target of rapamycin; CNI, calcineurin inhibitor; ESRD, end stage renal disease; AU, arbitrary units; GMT, geometric mean titer; ELISA, enzyme-linked immunoassay; IQR, interquartile range; NA, not available.

## Data Availability

The authors confirm that the data supporting the findings of this study are available within the article and its Appendix A.

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
