# Peer review of "A Systematic Review and Meta-Analysis of Serologic Response following Coronavirus Disease 2019 (COVID-19) Vaccination in Solid Organ Transplant Recipients"

_viruses, 2022, doi:10.3390/v14081822_

Round 1

Reviewer 1 Report

In this study, the authors have analyzed serologic response following coronavirus disease 2019 (COVID-19) vaccination in solid organ transplant (SOT) recipients including 44 observational studies with 6158 SOT recipients. Most studies were of mRNA vaccination (mRNA-1273 or BNT162b2). Their results suggest that a significant proportion of solid organ transplant recipients are unable to achieve a sufficient serologic response after completing not only the two series of vaccination but also the third booster dose. Thus, it is urgent to develop strategies for prophylaxis including modified vaccine schedules or the use of monoclonal antibodies in this vulnerable patient population.

Several suggestions:

1.      Lines 228-235, it is better to calculate the serologic response rates (percentages) of SOT and control subjects.

2.      Lines 239-249, it is better to calculate the serologic response rates (percentages) of SOT and control subjects.

3.      It is better to cite the references of the serologic response rates (percentages) of control subjects after a single dose, two doses and three doses. Then, it is easier to understand why 8.6%, 34.2%, 65.6% of SOT subjects are relatively low.

4.      It is better to cite the references after the following sentences: [COVID-19 vaccines were developed and a large number of studies demonstrated that mRNA COVID-19 vaccinations are efficacious and safe (line 60)]; [Guideline recommendations are that transplant recipients should receive COVID-19 vaccination unless there is any contraindication (line 64)]; [Most studies assessing the efficacy of COVID-19 vaccinations in transplant recipients are of limited sample size (line 68).]

Author Response

In this study, the authors have analyzed serologic response following coronavirus disease 2019 (COVID-19) vaccination in solid organ transplant (SOT) recipients including 44 observational studies with 6158 SOT recipients. Most studies were of mRNA vaccination (mRNA-1273 or BNT162b2). Their results suggest that a significant proportion of solid organ transplant recipients are unable to achieve a sufficient serologic response after completing not only the two series of vaccination but also the third booster dose. Thus, it is urgent to develop strategies for prophylaxis including modified vaccine schedules or the use of monoclonal antibodies in this vulnerable patient population.

Several suggestions:

  1. Lines 228-235, it is better to calculate the serologic response rates (percentages) of SOT and control subjects.

RESPONSE: Thank you for your comments. We have added the percentage of patients achieving a response (# of pts achieving a response / total # of pts).

  1. Lines 239-249, it is better to calculate the serologic response rates (percentages) of SOT and control subjects.

RESPONSE: Thank you for your comments. We have added the percentage of patients achieving a response (# of pts achieving a response / total # of pts).

  1. It is better to cite the references of the serologic response rates (percentages) of control subjects after a single dose, two doses and three doses. Then, it is easier to understand why 8.6%, 34.2%, 65.6% of SOT subjects are relatively low.

RESPONSE: Thank you for your comments. We had included this information in the Discussion section, but the addition of the percentages in the studies  comparing rates to controls would also make it easier to interpret these low rates.

  1. It is better to cite the references after the following sentences: [COVID-19 vaccines were developed and a large number of studies demonstrated that mRNA COVID-19 vaccinations are efficacious and safe (line 60)]; [Guideline recommendations are that transplant recipients should receive COVID-19 vaccination unless there is any contraindication (line 64)]; [Most studies assessing the efficacy of COVID-19 vaccinations in transplant recipients are of limited sample size (line 68).]

RESPONSE: Thank you for your comments. We have added references to these sentences.

Reviewer 2 Report

The presented research results constitute an important contribution to the current state of knowledge on the possibilities of COVID 19 prophylaxis in people after solid organ transplantation. It is an analysis of the existing knowledge, mainly based on available observational and cohort studies on this subject. The methodology was presented correctly and comprehensively. In the part describing the obtained results, the authors did not avoid a few minor errors, mainly editorial. For example, in lines 198 and 199, the words are combined into one string, with no spaces. Similarly in lines 226 and 227.

In addition, no answer to the important question - if the patients did not convert after vaccination, it had consequences in the increased number of COVID 19 cases. by doctors caring for these people in transplant clinics.

Author Response

The presented research results constitute an important contribution to the current state of knowledge on the possibilities of COVID 19 prophylaxis in people after solid organ transplantation. It is an analysis of the existing knowledge, mainly based on available observational and cohort studies on this subject. The methodology was presented correctly and comprehensively. In the part describing the obtained results, the authors did not avoid a few minor errors, mainly editorial. For example, in lines 198 and 199, the words are combined into one string, with no spaces. Similarly in lines 226 and 227.

RESPONSE: Thank you for your comments. We have corrected the parts where the sentences were connected in one string.

In addition, no answer to the important question - if the patients did not convert after vaccination, it had consequences in the increased number of COVID 19 cases. by doctors caring for these people in transplant clinics.

RESPONSE: Thank you for your comments. This is an important issues, but we were unable to answer it in our meta-analysis. We have included this in the limitations. “Further studies assessing whether incomplete serologic response to vaccination would prevent symptomatic or severe COVID-19 in SOT patients are also warranted”.

Reviewer 3 Report

This article is very well presented and has focused on how immune-microenvironment in solid organ transplant patients affects the Covid vaccine strategy that should not be overlooked. I have the following concerns about the article,

1. The author has presented the data of patients who have been vaccinated but if would be necessary to explain whether there were any patients exposed to Covid infection post vaccination.

2. Are there any patients who had undergone immunosuppressive therapy required for organ transplant?

3. Also the author should include a table explaining the ages of the patients included in the study which may affect the serum antibody titer.

Author Response

This article is very well presented and has focused on how immune-microenvironment in solid organ transplant patients affects the Covid vaccine strategy that should not be overlooked. I have the following concerns about the article,

  1. The author has presented the data of patients who have been vaccinated but if would be necessary to explain whether there were any patients exposed to Covid infection post vaccination.

RESPONSE: Thank you for your comments. This is an important issues, but we were unable to answer it in our meta-analysis. We have included this in the limitations. “Further studies assessing whether incomplete serologic response to vaccination would prevent symptomatic or severe COVID-19 in SOT patients are also warranted”.

  1. Are there any patients who had undergone immunosuppressive therapy required for organ transplant?

RESPONSE: A majority of patients were taking immunosuppressive medications. This information is included in Table 1.

  1. Also the author should include a table explaining the ages of the patients included in the study which may affect the serum antibody titer.

RESPONSE: The information of age is included in Table 1.